# Effect of Web Flexibility in Gear Engagement: A Proposal of Analysis Strategy

**Fabio Bruzzone** * and **Carlo Rosso**

Department of Mechanical and Aerospace Engineering, Politecnico di Torino, 10139 Turin, Italy; carlo.rosso@polito.it
* Correspondence: fabio.bruzzone@polito.it

**Abstract:** Increasing torque and power density in geared transmissions is a constant trend, especially due to the electrification and the research of higher efficiency. Thinner web of the gears has led to potentially higher vibration amplitudes and noise levels, which need to be accounted for in the design stage to avoid fatigue or NVH problems. In the present paper a novel model to represent the dynamic interaction in geared transmissions is presented. The compliances of the shafts are taken into account in two different ways: in a simple strategy, by means of three-dimensional beam elements; in a more detailed methodology, by reduction in a FE model of the gear and shaft assembly. The different gears are connected using kinematic relationships exploited by means of rigid joints and rigid body elements. The flexibility of the teeth and of the gear body is introduced based on an established nonlinear calculation model and is employed as the only dynamic excitation source as a time varying mesh stiffness. Thanks to the reduced size of the matrices a direct integration scheme is used for the time domain analysis of the transmission. Such methodology enables the possibility of modeling dynamic contact loss and torque variations at different rotational velocities with reduced computational times in comparison to other approaches. The technique is then applied to two different geometries of driven gear: one with solid web and one with flexible web. The dynamic behavior of the two different solutions is studied and the differences are highlighted. The proposed approach proves to be more efficient than traditional multibody analyses and quicker than finite element approaches, while maintaining a similar accuracy. It proves to be effective also for studying the transmission with flexible web.

**Keywords:** gears; transmissions; dynamics; dynamic error

## 1. Introduction

One of the most ancient methods to transmit torque in machines is the use of gears: the possibility to modulate torque and speed, to delivery power to several users starting from a unique motor, the high power density elaborated by such mechanical solution are some of the biggest pros of gears. For those reasons gears are strongly used in the electrification of the automotive industry, in the wind energy technology and in aerospace power transmission. The increase in power supply and the necessity to reach higher efficiency makes the gears prone to higher probability of damage. Among damage mechanisms in gears, surface wear, noise and harshness are the most important and could lead to fatigue and crack propagation in the rim and web [1,2]. They are principally related to dynamic phenomena, which become even more important as the rotational speed increases. For those reasons a wide literature is available to study dynamic behavior of gear engagement both numerically and experimentally ([3–6]). The most important parameter introduced in such studies is the Dynamic Factor (DF), which is the ratio between the loads in dynamic conditions with respect to the nominal one [7]. Harris [8] was the first researcher that introduced the Static Transmission Error (STE), defined as the variation in stiffness during the mesh cycle of the engaged gear pairs. He highlighted also the importance of the STE in

generating dynamic overloads in gear meshing, which was also experimentally confirmed for pinion-gear [9,10] and multi-mesh configurations [11]. Several models of increased complexities were developed in literature, starting from the 1D lumped element models where the STE is directly used to compute the Dynamic Transmission Error (DTE) ([7,12,13]) while in other papers the STE is expressed as a sinusoidal function and coupled with torque variation with different frequencies introducing also a non-linear elastic contribution term, which can vanish due to backlash between the flanks [14]. Other authors did not consider the external fluctuations and they only considered excitation source coming from the Time Varying Mesh Stiffness (TVMS), which is computed from the STE at different torque levels [15]. In [16] an improved model including some degree of compliance of shafts and clearances in the bearings was proposed and in [17] an experimental verification of the non-linear jump phenomenon predicted by the numerical model was carried out. The TVMS was also used in analytical and numerical methodologies to study the influence of damping and backlash on possible chaotic phenomena in gears [18] also including the non-linear influence of journal bearings and the rotordynamic behavior of the shafts [19,20]. A great number of methods based on Finite Elements (FE) were proposed ([15,21–25]) with different degrees of details in the contact mechanics study, excitation loads and outputs, and some of these approaches were applied to planetary systems ([26–28]). However, the scope of those models is very limited since they are all based on some assumptions of applicability, for example, neglecting some flexibilities or limiting the suitableness of the model to occurrences where the main meshing frequency is far from the flexible modes of the shafts. Other cited models overcome those limitations, however, they are in turn extremely computationally intensive making them unsuitable to be used in the design phase or if the vibration suppression is not critical. The present paper deals with the possibility to create the basis of a dynamic model open to further improvements and additions. Based on standard and well-known FE joints, the engagement kinematics of a gear will be described as well as the coupling with another geared shaft. The element that connects the two gears is depicted as a spring with a stiffness constant computed based on the TVMS. In such a way, the unique dynamic excitation introduced is the mesh stiffness variation. The compliance of gear web is introduced by means of a reduced model. An example of a single stage transmission with two different driven gear geometries will be then detailed and studied in the time domain using a fast Newmark direct time integration scheme, in which all the main flexibilities and aspects of the engagement are correctly simulated. Finally, conclusions are drawn, and several further developments of this first approach will be detailed for future research.

## 2. Materials and Methods

### 2.1. Model Description

2.1.1. Rigid Gear Body

The dynamic model of a transmission is characterized by several elements. The more important elements to be described are the shafts, on which the gears are mounted. In the proposed approach those elements will be, in a first attempt, modelled as Timoshenko beam elements [29] with six Degrees Of Freedom ($DOF\ x, y, z, \theta_x, \theta_y, \theta_z$). Mass and stiffness matrices, $\boldsymbol{M_s}$ and $\boldsymbol{K_s}$, respectively, are built depending on the geometry and the discretization of the system. In such a way, each shaft will be connected to at least one gear and a gear element will be defined. This last will serve as a connection to the other shafts becoming the kinematic link between shafts according to the procedure herein described. The pinion gear, in the gear pair, is represented by means of a node $O_1$ belonging to the shaft beam elements and this node represents the gear location in the model.

According to gear theory [30], notable points $T_1$ and $C$ can be easily defined as the tangent point between the Line Of Action (LOA) and the base circle of the gear, and the pitch point, respectively, as visible in Figure 1. At point $C$, a virtual node is generated while two coincident virtual nodes ($T_{1,1}$ and $T_{1,2}$) are added at point $T_1$. Nodes $T_{1,1}$, $O_1$ and $T_{1,2}$,

*C* are connected by means of a Rigid Body Element (RBE) in a master-slave relationship. The master node DOF are collected in a transformation matrix as shown in Equation (1)

$$
\left\{ \begin{array}{c} u_m \\ u_s \end{array} \right\} = T_{rbe,i} u_m = \begin{bmatrix} 1 & & & & & & & & \\ & 1 & & & & & & & \\ & & 1 & & & & & & \\ & & & 1 & & & & & \\ & & & & 1 & & & & \\ & & & & & 1 & & & \\ 1 & & & 0 & -\Delta z & +\Delta y & & & \\ & 1 & & +\Delta z & 0 & -\Delta x & & & \\ & & 1 & -\Delta y & +\Delta x & 0 & & & \\ & & & 1 & & & & & \\ & & & & 1 & & & & \\ & & & & & \mathbf{1} & & & \end{bmatrix} u_m \tag{1}
$$

where $u_m$ indicates the displacements of the master nodes, while $u_s$ the slave's DOF. $\Delta x$, $\Delta y$ and $\Delta z$ represent the geometrical distance between the nodes according to the reference frame. Two of those transformation matrices are needed, which are assembled in the total transformation matrix $T_{RBE}$ using standard procedures and the kinematic chain is closed by joining nodes $T_{1,2}$, $T_{1,1}$ in a master slave relationship through a Rigid Joint (RJ) rigidly connecting their degrees of freedom creating a generalized displacement equality, while keeping the rotations uncoupled. This is achieved through a transformation matrix of the form

$$
\left\{ \begin{array}{c} u_m \\ u_s \end{array} \right\} = T_{RJ} u_m = \begin{bmatrix} & & & I & & & & & \\ & 1 & 0 & 0 & 0 & 0 & 0 & & \\ \mathbf{0} & 0 & 1 & 0 & 0 & 0 & 0 & \mathbf{0} \\ & 0 & 0 & 1 & 0 & 0 & 0 & & \end{bmatrix} u_m \tag{2}
$$

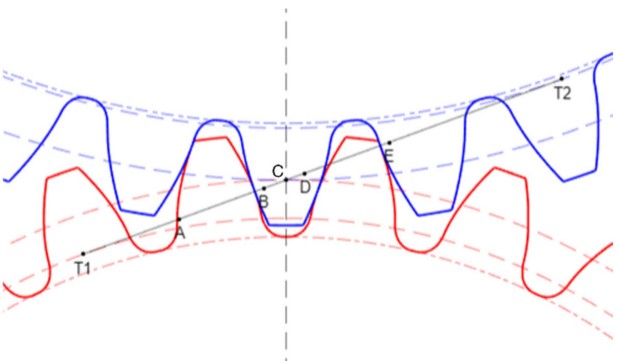

**Figure 1.** Location of points $C$, $T_1$ and $T_2$ during the engagement.

The final matrices are then obtained as

$$
M_i = T^T M_s T, \quad K_i = T^T K_s T \tag{3}
$$

where $T$ is the total transformation matrix obtained by $T = T_{RBE} T_{RJ}$. This process is repeated for the driven gear, simply considering points $O_2$ and $T_2$ and the connection chain is visualized in Figure 2. The shaft matrices of all the stages treated as previously described are then assembled in the global system matrices $M_z$ and $K_z$ using standard FE procedures [31]. To connect the different gear pairs on the different shafts, springs simulating the TVMS are positioned in point C thus closing the kinematic chain and connecting the different shafts. The method that will be described next is linear at each time instant, although it uses the results of the previous one to define the load and the value of the TVMS, thus making it globally nonlinear since the stiffness value varies depending on the applied load.

A linearized stiffness $k_{n,z}$ is estimated by TVMS, which is dependent on the instantaneous torque and position, and the element stiffness matrix $k_{c,n}$ is defined for the $n$th gear pair at the $z$th angular position as

$$k_{c,n} = k_{n,z} \begin{bmatrix} \sin(\alpha) & 0 & 0 & 0 & 0 & 0 \\ 0 & \cos(\alpha)\cos(\beta) & 0 & 0 & 0 & 0 \\ 0 & 0 & \cos(\alpha)\sin(\beta) & 0 & 0 & 0 \\ 0 & 0 & 0 & 0 & 0 & 0 \\ 0 & 0 & 0 & 0 & 0 & 0 \\ 0 & 0 & 0 & 0 & 0 & 0 \end{bmatrix} \qquad (4)$$

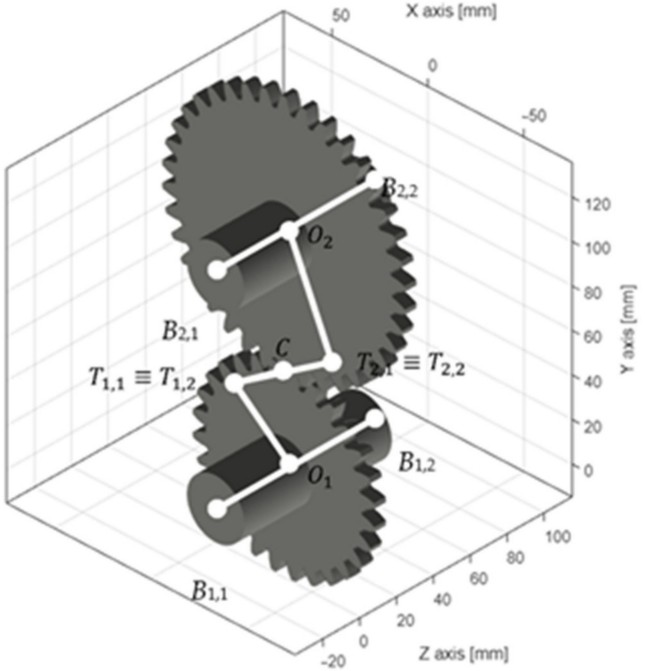

**Figure 2.** Scheme of the kinematic arrangement. FE elements schematized in white.

Which is then assembled in the global connection matrix for the entire system as [31]

$$K_{c,i} = \begin{bmatrix} 0 & & & & & & 0 \\ & \ddots & & & & & \\ & & k_{c,n} & \cdots & -k_{c,n} & & \\ \vdots & & \vdots & \ddots & \vdots & & \vdots \\ & & -k_{c,n} & \cdots & k_{c,n} & & \\ & & & & & \ddots & \\ 0 & & & \cdots & & & 0 \end{bmatrix} \qquad (5)$$

The obtained matrix is naturally time-dependent since the position, and hence the connection stiffness changes with time, and is computed for a discrete number of angular positions of the gears along a mesh cycle of the engagement. During the time integration process the instantaneous angular position will be estimated and the corresponding $k_{n,z}$ will be evaluated to form the corresponding $k_{c,n}$ matrix. In order to obtain such value, a weighted average between the two closest discrete values is performed. The value of $k_{n,z}$ is computed as

$$k_{n,z} = \frac{T}{r_b \cdot STE} \qquad (6)$$

where the $STE$ is computed using the code GeDy TrAss as described in [32] and $r_b$ is the base radius. The damping model chosen is the classical proportional Rayleigh damping $C_z$

$$C_z = \alpha_c M_z + \beta_c K_z \tag{7}$$

where the damping constants $\alpha_c$ and $\beta_c$ for the purposes of this paper are set to $3 \times 10^{-3}$ and $1.5 \times 10^{-8}$, respectively.

Once the assembly process is closed, an iterative integration scheme is performed. Newmark [33] scheme with constant acceleration is adopted. The constants used are

$$\alpha_{NM} = \frac{1}{4}, \; \delta_{NM} = \frac{1}{2} \tag{8}$$

In the Newmark integration scheme, by regrouping the terms of motion equation, the following expression is evaluated for the $i$th time step

$$\ddot{u}^*_i = S_i \delta r_i \tag{9}$$

where

$$S_i = M_z + \Delta t_i C_z + \Delta t_i^2 \alpha_{NM}(K_z + K_{c,i}) \tag{10}$$

where the time interval $\Delta t_i$ is kept constant. The residual vector $\delta r_i$ is obtained from the following matrices

$$\begin{aligned} D_i &= -(K_z + K_{c,i}) \\ V_i &= -C_z - \Delta t_i(K_z + K_{c,i}) \\ A_i &= -C_z(1 - \delta_{NM})\Delta t_i - (K_z + K_{c,i})\left(\tfrac{1}{2} - \alpha_{NM}\right)\Delta t_i^2 \end{aligned} \tag{11}$$

which are then assembled as

$$\delta r_i = f + D_i u_{i-1} + V_i \dot{u}_{i-1} + A_i \ddot{u}_{i-1} \tag{12}$$

As the Newmark scheme requires, $u_{i-1}$, $\dot{u}_{i-1}$, $\ddot{u}_{i-1}$ are, respectively, the displacements, velocities, and accelerations vectors at the previous time step. The vector $f$, that depicts the external forces, will be assumed constant in the single time instant and the unique value different from zero will be the torque $T$ applied to the $\theta_z$ DOF of the input shaft. Following the Newmark scheme, the displacements, velocities, and accelerations of the current step can be computed using the values obtained to the previous $i$-1 time step, namely for the velocities

$$\dot{u}^*_i = \dot{u}_{i-1} + (1 - \delta_{NM}M)\ddot{u}_{i-1} + \delta_{NM}\ddot{u}^*_i \Delta t_i \tag{13}$$

and for the displacements

$$u^*_i = u_{i-1} + \Delta t_i \dot{u}_{i-1} + \Delta t_i^2 \left(\frac{1}{2} - \alpha_{NM}\right)\ddot{u}_{i-1} + \alpha_{NM}\ddot{u}^*_i \Delta t_i^2 \tag{14}$$

Formally, the displacements, velocities and accelerations $u^*_i$, $\dot{u}^*_i$, $\ddot{u}^*_i$ thus obtained are marked by the apex $*$ due to a contact stiffness value being used for the corresponding angular position along the mesh cycle at the $i$th time instant. This stiffness value changes as the next angular position corresponding to $(i+1)$th time step, so the stiffness matrix of the model is modified accordingly. Using such a scheme, the torque change in time can also be considered, as a matter of fact the dynamic torque is not constant and an approach similar to the one shown in [34] is introduced. Several STE are computed at different torque levels in order to create a sort of look-up table, in which, by entering with torque and meshing position, an interpolated value of STE can be evaluated. In such a way, based on the dynamic value of dynamic torque at timestep $i - 1$ the correct value of TVMS $k_{n,z}$ is

computed by interpolation making the approach inherently nonlinear since it is position and load dependent.

### 2.1.2. Flexible Gear Body

If a solid gear is considered, no important effect can be predicted from the web of the gear, although if a thin web is present, it is expected to have a relevant effect on the dynamic behavior of the system.

It is not easy to take into account the web flexibility ([35–37]), however in the present paper, a possible solution for evaluating some of the effects is proposed.

The developed algorithm above described can be adapted to the presence of a flexible web. By means of a FE model of the gear and shaft body, the reduced model can be extracted. The proposed algorithm schematizes the shaft and the gear by means of three nodes placed on the axis of rotation (see Figure 2), two nodes represent the bearing connections ($B_{2,1}$ and $B_{2,2}$ in this example) to the shaft and one the gear ($O_2$ as described earlier). A rigid element chain connects the gear node to the engagement stiffness in order to simulate the gear coupling. Following this philosophy, the FE model of gear and shaft assembly is statically reduced to three nodes that represent the bearing connections and the link of the gear to coupling. The latter, in this paper, is considered to lay on the base circle ([10,38,39]). So, the FE model of the gear and shaft assembly is provided with three centerline additional nodes that correspond to the nodes $B_{2,1}$, $B_{2,2}$ and $O_2$. Three rigid elements (RBE2) connect the peripherical nodes (slaves) on the bearing locations and on the base circle to those three additional nodes. A Guyan reduction [40] is then performed using $B_{2,1}$, $B_{2,2}$ and $O_2$ as master nodes and the reduced stiffness and mass matrices are estimated. In Figure 3, the FE model of a gear and shaft assembly is shown, which will be used as comparison to evaluate the effectiveness of the proposed methodology.

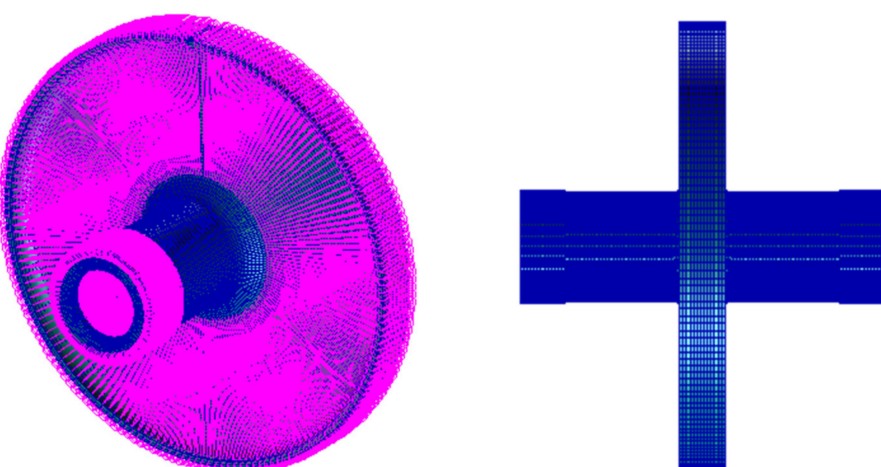

**Figure 3.** FE model of a gear and shaft assembly, in purple the 3 RBE2 that connect the peripherical nodes of the bearing locations and base circle to the master nodes.

### 2.2. Case Study

The proposed methodology is applied to a pair of gears, as depicted in Table 1, with a simple shaft geometry. Pinion and gear are mounted on a shaft with constant diameter and in the middle of the span. Bearings are connected to the two free ends of the shaft. For pinion and gear the shaft has the same length, however, in the case of the pinion it is solid, and its diameter is 36 mm. For the gear, a hollow shaft is considered, the features of which are reported in Table 1.

**Table 1.** Test transmission main parameters.

| | Thick Web | | Thin Web | |
|---|---|---|---|---|
| | **Pinion** | **Gear** | **Pinion** | **Gear** |
| Number of teeth $z$ $[-]$ | 19 | 100 | 19 | 100 |
| Module $m$ [mm] | 2.625 | | 2.625 | |
| Pressure angle $\alpha$ [°] | 20 | | 20 | |
| Helix angle $\beta$ [°] | 0 | | 0 | |
| Facewidth $b$ [mm] | 25 | | 25 | |
| Web thickness [mm] | constant | 25 | variable | 4 to 8 |
| Shaft diameters $D$ ext, $d$ int [mm] | 60 | 40 | 60 | 40 |
| Shaft length $L$ [mm] | 200 | | | |
| | Material: Steel | | | |
| Young's modulus $E$ [MPa] | 200,000 | | | |
| Poisson coefficient $\nu$ $[-]$ | 0.3 | | | |
| Density $\rho$ $\left[\frac{\text{kg}}{\text{m}^3}\right]$ | 7800 | | | |

The pinion shaft is modelled by means of Timoshenko beams (as described in Section 2.1.1) whereas the driven gear is represented by $18 \times 18$ stiffness and mass matrices as described in Section 2.1.2. Two different driven gear geometries are also considered, one with solid web and one with thin web. In Table 1 and in Figure 4 the two driven gear geometries are shown.

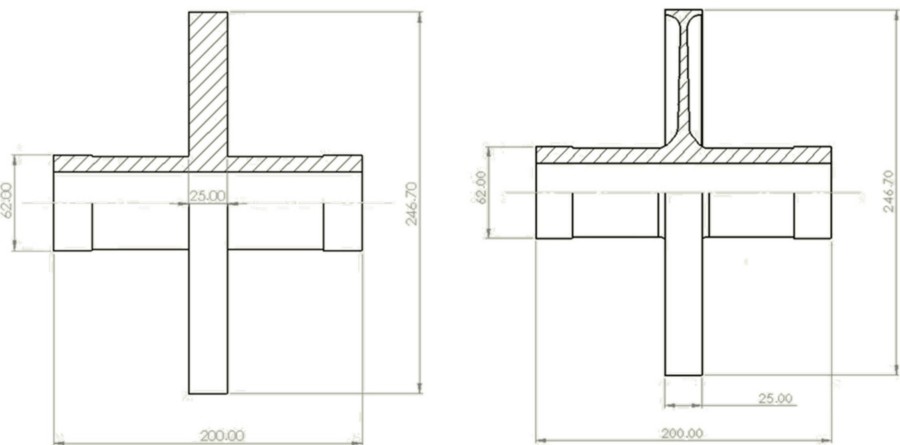

**Figure 4.** Drawing of the driven gear: on the left solid geometry, on the right flexible web geometry.

### 3. Results

Two models are then prepared in GeDy TrAss software, one concerning the pair pinion solid driven gear and one for pinion and flexible web gear obtained as reduction in the FE model shown in Figure 3. A constant torque of $T = 250$ Nm is applied on one end of the pinion shaft, the bearings are considered as displacement constraints in all the linear direction, whereas the rotations are all free. Only one end of the driven shaft has a locked rotation in order to simulate the torque constraint. In Figure 5, the model on GeDy TrAss software is presented.

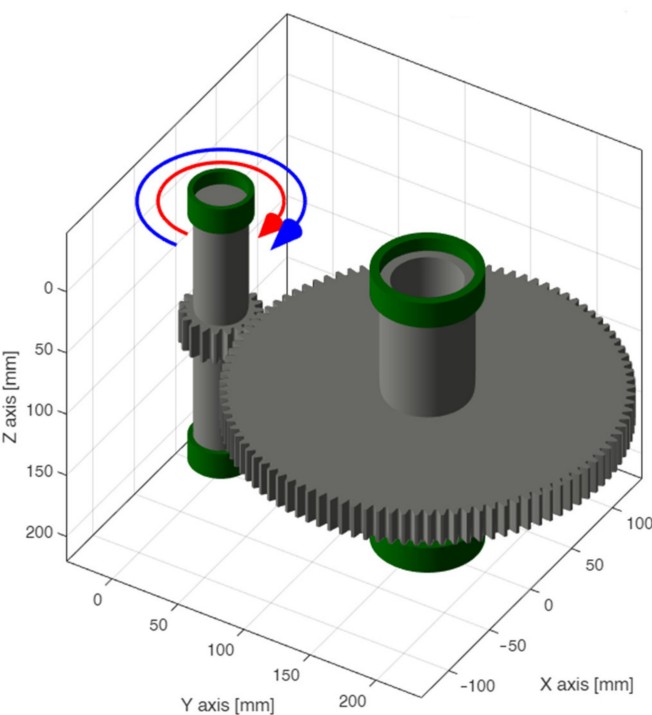

**Figure 5.** Test transmission model in the GeDy TrAss software.

A 5 s speed-up (1000 ÷ 10,000 RPM) will be simulated under those conditions with constant acceleration under a constant timestep equal to $10^{-5}$ s. Displacements, velocities, accelerations, and forces exchanged are registered for the entire simulation time at a few key points, namely the constraints and the mesh points. The different STE computed for the test cases are visible in Figure 6, which are used to compute the TVMS as previously described. As depicted in Figure 6, the static transmission error is affected by the web of the driven gear. A higher mean value of the error for the flexible web and a different trend of the Peak-to-Peak (P2P) can be highlighted. It is expected that these aspects will affect dynamic behavior. The quasi-static method [25] used to compute the STE and the P2P STE considers in the calculation more teeth than are needed in order to be able to evaluate an extremely early onset of contact or late disengagement, which does not happen in the present case and for this reason the results Teeth 1 and 5 are not visible in Figure 6a,b since they are never in contact.

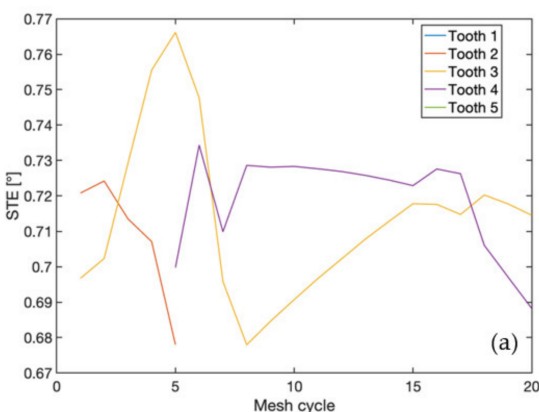
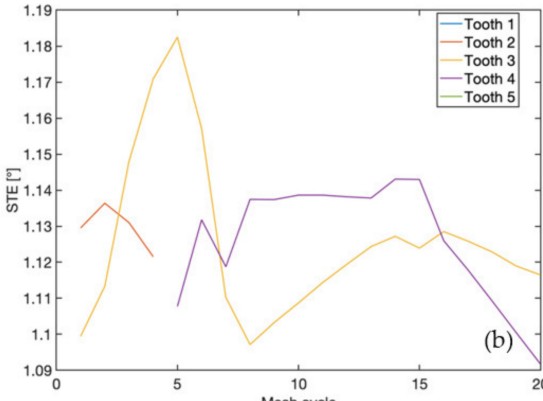

**Figure 6.** *Cont.*

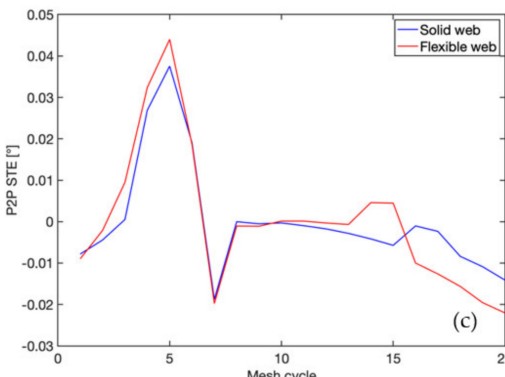

**Figure 6.** STE evaluation for the solid web (**a**), flexible web (**b**) and the comparison of the peak-to-peak STE between the two solutions (**c**).

In Figure 7 the dynamic variation in torque is reported. It is evident that the different geometry of the web affects the results. The maximum amplitude is almost the same, however the frequency and the velocity of the maximum is different from the two solutions. Similar considerations can be extracted from the dynamic forces experienced by the constraints during the time simulation. As visible in Figure 8, the dynamic force in the radial direction on the constraint $B_{1,1}$ is similar for the two solutions, however, the amplitudes of the oscillation of the force are generally higher when the thin web is adopted.

The proposed methodology is also able to provide the time histories of all the displacements, velocities and accelerations, so spectrograms [41] can be derived to highlight possible resonances using the Short Time Fourier Transform (STFT) [42].

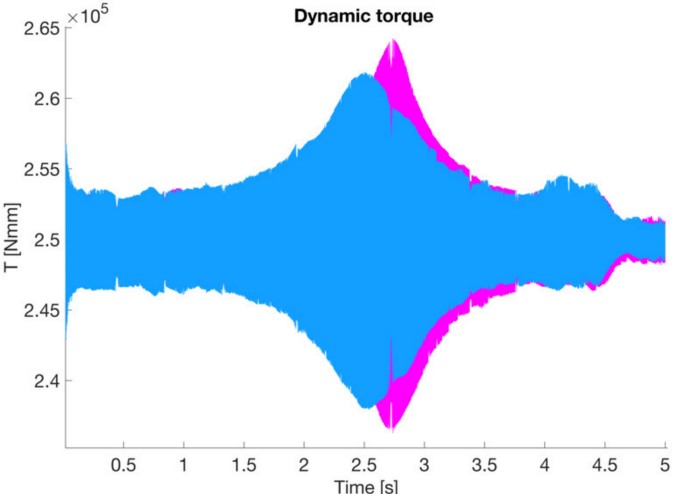

**Figure 7.** Dynamic variation in the torque for the two different geometries (blue—solid web, purple—flexible web).

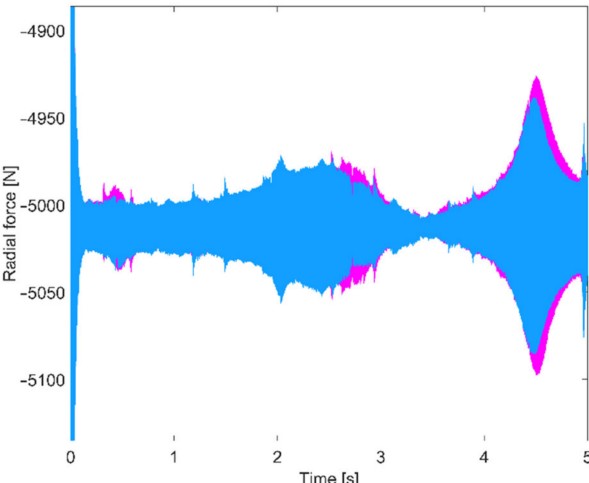

**Figure 8.** Dynamic variation in the radial force for the two different geometries on $B_{1,1}$ (blue—solid web, purple—flexible web).

In Figure 9 the spectrograms of the tangential displacement of node 1, where torque is applied on the pinion shaft, are displayed for both the geometries. The different behavior of the two geometries is quite evident. The flexible web presents a crossing between a natural frequency of the transmission and the fourth harmonics of the gear meshing frequency (usually, the meshing frequency is computed by multiplying the rotating frequency of the shaft by the number of teeth of the gear as $z_i\Omega$). It can be appreciated that this intersection presents higher amplitude values than that of the solid web. Similar reasoning can be deployed by taking into account the torsional degree of freedom. In Figure 10 the comparison of the two spectrograms for the rotation around the rotation axis for solid and flexible web systems is reported. It is evident that the flexible web system experiences higher amplitude response to the dynamic excitation due to engagement.

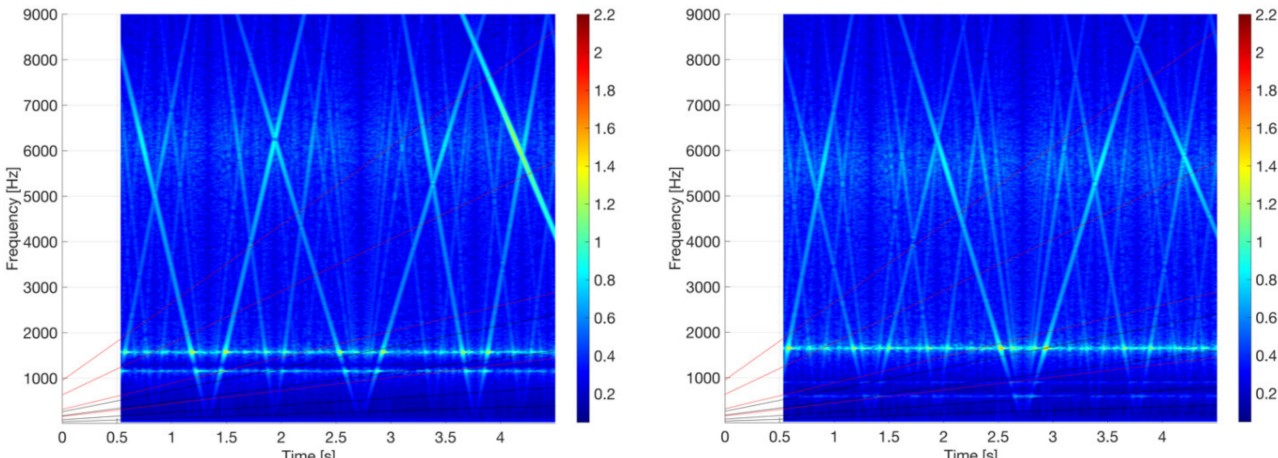

**Figure 9.** Spectrogram of tangential displacement on node 1 on the left for the solid web, on the right for the flexible web. The red and black sloped lines represent the harmonics of the meshing frequency of, respectively, pinion (red) and gear (black). The yellow circle indicates the crossing between a natural frequency and a harmonic of meshing frequency. Colormap reports the amplitude of the response and the unit is $\sqrt{\sqrt{mm}}$, this due to the little value of the displacement.

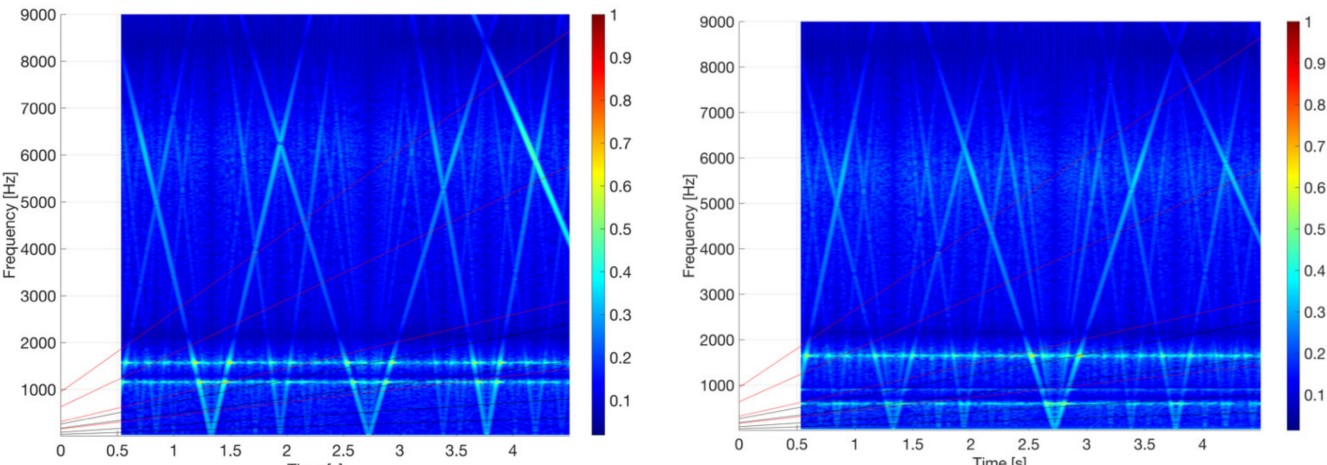

**Figure 10.** Spectrogram of torsional displacement on node 1 on the left for the solid web, on the right for the flexible web. The red and black sloped lines represent the harmonics of the meshing frequency of, respectively, pinion (red) and gear (black). The yellow circle indicates the crossing between a natural frequency and a harmonic of meshing frequency. Colormap reports the amplitude of the response and the unit is $\sqrt{\sqrt{\mathrm{mm}}}$, this due to the little value of the displacement.

## 4. Conclusions

The proposed methodology proves to be an accurate and fast alternative to a more complex model, such as co-simulation with multibody and FEM. The accuracy is given by the reduction procedure and the quickness is guaranteed by the Newmark integration scheme. In the particular case herein studied, two different geometries of the same driven gear are used to test the methodology. As expected, both STE and Dynamic Factor are different in the two cases, so this technique proves to be capable of appreciating such differences. Indeed, the main resonant frequencies of the two different models for the presented case do not show ample differences, however, due to the increased flexibility when the web is thin, an increase in the mean value of the STE is found as visible in Figure 6 while the total variation remains similar. The increased flexibility of the system also contributes to the excitation of the engagement, thus also incrementing the response in terms of dynamic torque and forces, which can be appreciated in Figures 7 and 8. The rigid web model also shows smaller dynamic displacements as shown by the spectrograms of Figures 9 and 10 as expected. By means of a static reduction, it is possible to consider the static stiffness of the web and not the flexibility due to ring in-plane and out-of-plane modes, however it is faster anyway than a FEM model solved in time domain and capable of providing some design indications. Indeed, the test cases shown here were computed in just over six minutes on a normal laptop computer (CPU Intel Core i7-7700 and 16Gb RAM), which means a huge time saving when compared to other methods, which require days of calculations and special hardware. In line of principle, the time required to solve those test cases is dependent on the square of the dimension of the matrices, which can be huge when FE are used, while it is kept to a minimum using this approach. As an estimate, the fastest flexible FE approach at the lowest accuracy settings needs around seven seconds to solve one time instant, while this method only needs a small fraction of a second with similar accuracy. Such methodology has great potential in the predesign phase, for pre-addressing some dynamic issues due to web flexibility. For future development the authors plan the introduction of a more complex reduction technique to also take into account the ring modes ([43]), as well as experimentally verifying the results by modifying a quasi-static test bench [44] for dynamic tests.

## 5. Patents

Some parts of the proposed technique are covered by patent It-102020000029570-02/12/2020 now under extension to European Agency.

**Author Contributions:** Conceptualization, F.B. and C.R.; methodology, F.B. and C.R.; software, F.B.; validation, F.B. and C.R.; writing—original draft preparation, F.B. and C.R.; writing—review and editing, F.B. and C.R. All authors have read and agreed to the published version of the manuscript.

**Funding:** This research received no external funding.

**Conflicts of Interest:** The authors declare no conflict of interest.

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
