# Peer review of "Effect of Web Flexibility in Gear Engagement: A Proposal of Analysis Strategy"

_vibration, doi:10.3390/vibration5020013_

Round 1

Reviewer 1 Report

The proposed paper on web flexibility in gear engagement was well-researched and presented, with both adequate and relevant references throughout the text. 

This research programme was a theoretical and mathemtical treatment and was based upon a simple spur gear train, utilising the common-placed and standard 20 degree pressure angled gears. It would be interesting in any further research-work, to expand this to the more complex heical/double helical gears, with varying pressure angles - but this is merely a suggestion?

The prospective treatment does deal with spur gears and their dynamic behaviour in operation, therefore should be published in my opinion.

Author Response

Dear Reviewer,

Thank you very much for your comments. Your suggestion is surely interesting, we plan to increase the complexity of the model step by step and surely helical gears are in our scope. The model is already capable of analyzing spur gears with different pressure angles, not only the standard ones, but non-standard as well. We chose this test case because it is one of the most commonly utilized.

Thank you again for you suggestions and comments.

Reviewer 2 Report

  1. The study results of the dynamic behavior of the two different solutions should be clearly indicated and  summarized briefly in section 5.
  2. The nonlinear calculation model should be described in detail.
  3. The  work should be verified by experiment.

Author Response

Dear Reviewer,

We thank you for your inputs and we have modified the paper accordingly. The results of the dynamic behavior of the two proposed solutions have been summarized in the conclusions (now numbered paragraph 4) and we hope that this clarifies some aspects. We have also extended the explanation of the calculation model, especially we believe we have clarified the nonlinear approach. The connection stiffness matrix changes at every timestep since the gear rotates and hence the contact point changes, altering the contact stiffness as well. Also the dynamic torque influences this parameter and at each timestep the correct value is utilized based on the interpolation of the lookup table of the STE at different loads computed in the setup phase. We plan to verify with experiments this approach, but building a dedicated test bench for this purpose takes time and money, so we hope to be able to publish this paper in the meantime. We also plan to improve the complexity of the model so we will be able to compare the experimental results when the model is more complete.

Thank you again for your suggestions.

Reviewer 3 Report

  1. The introduction does not point out the necessity of conducting research on this issue ; although the introduction cited a large number of literature, but did not point out the shortcomings of previous studies and the content of this study compared with what advantages, please specify.
  2. The theoretical derivation is not rigorous, and the author ' s innovation points are not found. The expressions (9) and (13) are inconsistent.

  1. The calculation results of examples are lack of comparison, and the accuracy of the method is not reflected.

  1. The authors point out that the new idea proposed in this paper is more accurate and fast than the finite element method in calculating complex models, and how to prove it ?

Author Response

Dear Reviewer,

Thank you for your suggestions, we have taken them in great consideration and we modified the paper according to your inputs.

In the introduction we specified lack of the other methods and why our model fills some of the needs. As stated more complicated models based of FE can take into account many flexibilities, however the computational times are enormous which makes them unfeasible to be utilized in the predesign phase since the time to market would be too long. Other models are simpler and faster, but many factors are not taken into consideration. Our approach includes several flexibilities, and the calculation can be run quickly, allowing fast analyses and highlighting possible resonances that can then be addressed.

The theoretical derivation has been corrected, the equations are now consistent.

Objective of the paper was to analyze the effect of including the flexibility of the web in the model and we hope that the additions that we included in the paper now clear this aspect.

As stated, some FE models are more accurate, but with much longer calculation times. For example, a commercial software that can take into account the change in contact stiffness during the engagement takes at least 7 seconds for each time instant at the lowest accuracy setting because the matrices are large. Our approach uses small matrices thanks to the beam elements simulating the shafts, and the contact is precomputed at different load levels, ensuring a good accuracy with much faster calculation times (only a fraction of a second for each timestep).

We hope that the modifications that we included in the paper now clear the doubts. Thank you very much.

Reviewer 4 Report

1 Please, explain in more detail the purpose of Figure 3. What is its difference from the standard finite element method

2 It is clearer to use the value of 250 Nm

3 In lines 223-225 You state about spectrograms and resonant frequencies. Is it possible to provide this for a better understanding and effectiveness of the calculation method?

4. In fig. 6a and 6b lack graphs for other teeth.

Author Response

Dear Reviewer,

Thank you for your comments and suggestions. We have included your inputs in the revised version of the paper and we hope that the modifications we made now make it acceptable for publication.

In Figure 3 the standard FE model of the shaft is shown, which is used after a reduction technique to compare the results of the dynamic analysis considering the web as flexible. In our model the web is modeled with rigid elements but using the depicted FE model we can compare the results of including the flexible web.

We modified the value of the input torque and specified some details on the spectrograms and how to obtain them from the results in the time domain starting from the displacements as shown in the paper. This argument is vast and we believe that further discussing this matter would be out of the scope of the present paper, but the references we included are a good starting point for this topic.

Thank you for pointing out the lack of the lines for teeth 1 and 5 in that figure. The quasi static model that we used to compute the STE at different loads can deal with early engagement and late contact loss between the teeth, so more teeth than needed are included in the simulation and results are shown only if there is contact in that teeth pair. In this case the contact ratio is above 2, so a minimum of 3 teeth pairs are needed. To be sure to capture every possible contact in the analysis also one pair before and one after the 3 main pairs is included, but in the present case they do not have any effect. This is why teeth pairs 1 and 5 are present in the legend, but missing from the graph. We hope that this explanation and the details included in the paper now clarifies this aspect.

Thank you again for your suggestions, we hope that the modifications implemented in the revised paper make it acceptable for publication.

Reviewer 5 Report

The comments are in the pdf document.

Author Response

Dear reviewer,

We thank you for your comments and suggestions. Below is a response to your indications that we included in the revised version of the paper.

  • We have improved the description of the model and clarified some aspects which were probably not so clear in the original form. The model is complete and working and can be applied to all cylindrical gears, but we plan to further improve it with time. The present test cases aimed at showing the influence of the flexibility of the web.
  • We have carefully revised the writing of the paper and have cited in the text Figures 1 and 2 as you pointed out for better use of the reader.
  • We improved the description of the model in the text and we now hope that this aspect is clear. Since the gears rotate the connection stiffness matrix changes at every timestep and hence the contact point changes, altering the contact stiffness as well. Also the dynamic torque influences this parameter and at each timestep the correct value is utilized based on the interpolation of the lookup table of the STE at different loads computed in the setup phase.
  • We have included more details in the analysis and summarized the differences in the conclusions.
  • We chose 9kHz as frequency upper limit for the spectrograms because no other major resonance above this value was visible. Including higher frequencies in the graphs would have made them unreadable, however the analysis cut-off frequency is 50kHz. The main orders of excitation are all shown and if the rotational velocity limit would be increased only the sub-harmonics of the engagement would cross with the modes frequencies, causing smaller resonances than the main one which is shown in the figures.
  • We have improved the explanation of the connection stiffness matrix in the text and we hope that this makes its meaning clear.
  • The value of the damping constants was listed in Table 1, but it has now been moved in the text part for better comprehension.
  • We misnumbered the paragraph, part four is now the conclusions.

Thank you very much for your valued comments and suggestions, we now hope that this work is publishable.

Round 2

Reviewer 2 Report

ACCEPTED

Reviewer 5 Report

please check the legend in Fig. 6 and 7.